# Wnt Signaling Inhibitors and Their Promising Role in Tumor Treatment

**DOI:** 10.3390/ijms24076733

**Published:** 2023-04-04

**Authors:** Nives Pećina-Šlaus, Sara Aničić, Anja Bukovac, Anja Kafka

**Affiliations:** 1Laboratory of Neuro-Oncology, Croatian Institute for Brain Research, School of Medicine, University of Zagreb, Šalata 12, 10000 Zagreb, Croatia; anja.bukovac@mef.hr (A.B.); anja.kafka@mef.hr (A.K.); 2Department of Biology, School of Medicine, University of Zagreb, Šalata 3, 10000 Zagreb, Croatia; 3Department of Physiology and Immunology, School of Medicine, University of Zagreb, Šalata 3, 10000 Zagreb, Croatia; saraanicic1996@gmail.com; 4Laboratory for Molecular Immunology, Croatian Institute for Brain Research, School of Medicine, University of Zagreb, 10000 Zagreb, Croatia

**Keywords:** Wnt signaling pathway, β-catenin, porcupine, Wnt inhibitors, mutations, tumors

## Abstract

In a continuous search for the improvement of antitumor therapies, the inhibition of the Wnt signaling pathway has been recognized as a promising target. The altered functioning of the Wnt signaling in human tumors points to the strategy of the inhibition of its activity that would impact the clinical outcomes and survival of patients. Because the Wnt pathway is often mutated or epigenetically altered in tumors, which promotes its activation, inhibitors of Wnt signaling are being intensively investigated. It has been shown that knocking down specific components of the Wnt pathway has inhibitory effects on tumor growth in vivo and in vitro. Thus, similar effects are expected from the application of Wnt inhibitors. In the last decades, molecules acting as inhibitors on the pathway’s specific molecular levels have been identified and characterized. This review will discuss the inhibitors of the canonical Wnt pathway, summarize knowledge on their effectiveness as therapeutics, and debate their side effects. The role of the components frequently mutated in various tumors that are principal targets for Wnt inhibitors is also going to be brought to the reader’s attention. Some of the molecules identified as Wnt pathway inhibitors have reached early stages of clinical trials, and some have only just been discovered. All things considered, inhibition of the Wnt signaling pathway shows potential for the development of future therapies.

## 1. Introduction

In the last thirty years, inhibitors of the Wnt signaling pathway have been identified and characterized along with the functional explanation of the pathway’s molecular targets [1,2,3]. Here, we aim to summarize the knowledge on Wnt inhibitors, discuss their effectiveness as therapeutics, and debate their side effects. However, before describing the relevant inhibitors and their targets, it is important to briefly address the mechanisms of Wnt signaling.

Wnt signaling is a conserved cellular pathway in all multicellular organisms that has been studied for more than four decades. The name was coined from the names of two genes, mouse *int-1* and Drosophila’s wingless (*wg*). The discovery of a novel cellular proto-oncogene *int-1*, which was later on mapped to the chromosomal position of Drosophila gene *wg*, launched the marvelous research on this essential pathway and its many important components.

It is generally accepted that Wnt signaling consists of canonical β-catenin and two non-canonical beta-catenin independent pathways—the planar cell polarity (PCP) and the Wnt/Ca^2+^ [4,5]. Canonical or classical Wnt signaling is involved in processes of body axes formation during development and, in the morphogenesis of limbs, the central nervous system, and other organs [6,7]. In adult organisms, its role mainly lies in stem cell regeneration, regulation of proliferation, and differentiation [8,9]. Activation of the β-catenin Wnt pathway leads to the transcription of the Wnt target genes. The planar cell polarity regulates the shaping of the cytoskeleton and the polarization of cells along the apical–basal plane, whereas the Wnt/Ca^2+^ pathway regulates cytoplasmic concentration of calcium ions through their exit from the endoplasmic reticulum [1,10,11].

The aberrant canonical Wnt pathway is involved in the formation and evolution of various types of tumors [12,13]. In addition, the PCP and Wnt/Ca^2+^ pathways are thought to be important in the acquisition of metastatic properties, primarily because of their role in cytoskeletal reorganization [14,15]. There are 19 different Wnt ligands [16] in mammals and humans that can activate different Wnt signaling branches, depending on receptors, coreceptors, and other regulatory molecules at a given time [17]. Thus, some Wnt ligands can activate both canonical and non-canonical pathways, whereas others act specifically. In the present review we will focus our attention on beta-catenin classical signaling because it is a major cellular and best-characterized branch, especially when inhibitors are in question.

## 2. The Canonical (β-Catenin Dependent) WNT Pathway

The canonical Wnt signaling pathway has been extensively characterized both in development and disease. When active, it contributes to the stabilization of cytoplasmic β-catenin, a molecule that in combination with transcription factors and coactivators TCF, LEF, PYGO, BCL9, p300, or CBP activates the transcription of Wnt target genes [18]. The first level of the canonical Wnt signaling is represented by Wnt ligands, secreted glycoproteins that undergo double palmitoylation by the porcupine protein [19,20] as well as glycosylation in the endoplasmic reticulum and Golgi, from where they are secreted into the extracellular space in secretory vesicles [21,22]. In the extracellular space, palmitoylated Wnt glycoproteins bind to Frizzled (FZD) receptors and LRP5/6 coreceptors, and the activation of Wnt signaling commences. Wnts can act as autocrine as well as paracrine signaling molecules [8,23]. Once a Wnt ligand binds to the receptor Frizzled [24], a complex forms at the membrane consisting of Wnt, FZD, and LRP5/6. Phosphorylated coreceptor LRP5/6 then induces intracellular signaling by recruiting the scaffold protein AXIN1/2. Next, after being hyperphosphorylated, another scaffold protein—Dishevelled (DVL)—is also recruited to the membrane complex. With its DIX domain, DVL binds both to the intracellular portion of the FZD receptor and to AXIN [25], and in such a manner, AXIN’s integration into the so-called β-catenin degradation complex is prevented, it is separated from it and β-catenin fails to degrade. Consequently, the cytoplasmic levels of β-catenin will increase, which leads to its translocation into the nucleus and subsequent transcriptional stimulation of the Wnt target genes [3,11] (Figure 1). Some of the target genes are involved in cell cycle regulation (*c-myc*, *N-myc*, *c-jun*, *cyclin D1*, *Sox9*) [26], some are involved in processes that facilitate metastatic cancer spread (*MMP-7*, *VEGF*), and the *TERT* gene encodes a component of the telomerase enzyme whose activity is an important feature of cellular immortality [27].

When the Wnt pathway is inactive, the β-catenin degradation complex, sometimes referred as degradosome, phosphorylates free cytoplasmic β-catenin, making it susceptible to ubiquitination and proteasome degradation [11,28,29]. Under normal circumstances, the degradation complex consists of AXIN, APC, casein kinase 1α (CK-1α), and glycogen synthase kinase 3β (GSK3β) [3,8,30,31,32]. CK-1α starts the phosphorylation of β-catenin on Ser45, allowing GSK3β to recognize this amino acid, and proceeds to the phosphorylation of serine/threonine residues at Thr41, Ser37, and Ser33 sites toward the N-amino end of β-catenin [33,34]. The N-terminal domain of β-catenin, which is nota bene targeted by the majority of mutations, contains phosphorylation sites for GSK3β, CK-1α, and Bcl9 [8,30,33]. On the other hand, the C-terminal domain of β-catenin serves to transactivate Wnt target genes through interaction with TCF/LEF transcription co-factors [35,36].

β-catenin transcriptional activity does not necessarily have the same consequences under all conditions nor in every cell type. To generate a transcriptionally active complex, β-catenin–TCF recruits the transcriptional co-activator CREB-binding protein (CBP) or its closely related homologue E1A-associated protein p300. The preference to the specific coactivator is the first step in deciding between programs of differentiation and proliferation. Thus, when considering the optimum therapeutic effect, inhibitors acting specifically on each of these two programs should be taken into consideration [1,37].

Because β-catenin does not contain nuclear localization signal sequences (NLS), the mechanism of its nuclear transfer is still debated. There are conflicting reports from several studies [38,39,40] on the issue of direct binding of β-catenin to the nucleoporin complex of the nuclear envelope. Therefore, inhibitors of the interaction of β-catenin and the nucleoporin complex might also have potential therapeutic benefits [41,42,43] (Figure 1).

It is important to understand that there is another pool of β-catenin in the cell. β-catenin is part of the protein complex that forms intercellular adherens junctions, along with E-cadherin on the outer side of the junction and α-catenin bound to the inner cytoskeletal proteins. Adherens junctions between epithelial cells preserve cell-to-cell adhesion but also prevent the dissociation of individual tumor cells from the tumor mass [44]. The structure of the β-catenin protein, which is 781 amino acids long in humans, is composed of a central region, where up of 12 armadillo repeats reside, and of N- and C-terminal domains [45,46,47]. A conserved Helix-C motif is located near the C-terminal domain. The N- and C-terminal domains can be structurally flexible, whereas the central region forms a scaffold on which many β-catenin binding proteins can attach. Through its domain of 12 armadillo repeats [36], β-catenin interacts with E-cadherin, APC, PYGO, axin, and partly with TCF. However, it has been shown that the binding sites for E-cadherin and TCF do not overlap completely, and armadillo repeats at different positions are required to bind β-catenin to TCF or E-cadherin. Namely, binding of β-catenin to TCF requires from 3 to 10 armadillo repeats, whereas binding to E-cadherin requires all 12 repeats [48,49]. In addition, beta-catenin’s C-terminal region can bind its own armadillo region and this conformation, sometimes referred to as closed, can lead to the selectivity of β-catenin binding to TCF [49,50]. Therefore, such conformation when the C-terminal domain overlaps part of the E-cadherin binding site regulates β-catenin selective binding to TCF [49]. It is hypothesized that activation of the Wnt pathway, besides increasing the amount of β-catenin in the cytoplasm, also causes a conformational change in β-catenin that favors TCF binding over E-cadherin. By inhibiting β-catenin sites for the interaction with TCF, Wnt target genes would be shut down. On the other hand, when intervening by deleting a part of the C-terminal domain helix C, the function of the central part would be preserved, whereas the transcriptional activity of β-catenin would be disabled [36]. It has also been reported that the interaction of α-catenin with the N-terminal domain of β-catenin leads to a conformational change of the C-terminal domain that facilitates the binding to E-cadherin [36,49,50]. Additionally, according to one study [51], protein kinase B (PKB/Akt) can cause β-catenin dissociation from intercellular attachments by phosphorylation at the Ser552, after which it accumulates in the cytosol and binds to the 14-3-3ζ protein that stimulates its translocation into the nucleus [43].

It is also important to know that there are six families of proteins that act as extracellular inhibitors of the Wnt pathway. These are Dickkopf, WIF, SOST/Sclerostin, Cerberus, SFRP, and IGFBP4 [52,53,54]. DKK1, DKK2, and Sclerostin bind to LRP5 and 6 and act as antagonists of the Wnt pathway, thus are often characterized as tumor suppressors [53]. The inhibitory action of the SFRP glycoprotein family, which includes SFRP1, 2, 3, 4, and 5, in humans is mediated through their two domains, the amino terminus Cysteine rich domain (CRD) and the carboxy-terminal Netrin-related motif (NTR). These domains allow SFRPs to bind to Wnt ligands, thus preventing them from binding to FZD receptors [53]. Additionally, SFRP proteins can form inactive complexes with the FZD receptor, which once again prevents Wnt activation [54].

The Wnt signaling pathway malfunctions in a number of tumors, including colorectal, breast, glioma, melanoma, pancreatic, and many others [17,55,56,57]. In addition, Wnt signaling plays a role in tumor invasiveness and metastatic spread because β-catenin forms adherens junctions that break down during cellular detachment [58,59]. β-catenin also activates the transcriptional repressors Slug and Snail, which reduce E-cadherin expression and thus promote epithelial mesenchymal transition (EMT) [60,61,62,63,64,65]. Therefore, mutations of the molecular components of the Wnt signaling also participate in EMT [27].

Changes in the molecules of the Wnt pathway in tumors can be mutational or non-mutational. Non-mutational changes include all levels of epigenetic changes—DNA methylation, histone modification, and RNA interference. Both type of changes promote tumor formation and invasiveness, which usually happens because of the loss of function of negative regulators or the overexpression of activators of the Wnt pathway. Generally, the most common Wnt pathway mutations in tumors are mutations of the *APC* and the *CTNNB1* genes [66]. The mutant APC protein lacks the ability to bind AXIN and to degrade β-catenin. A mutation in exon 3 of the *CTNNB1* gene, which encodes β-catenin, accounts for 90% of mutations, leading to a change in the N-terminal domain whereby β-catenin loses its key sites for degradation complex enzyme activity [65].

Historically, the first Wnt mutations were detected in sporadic as well as hereditary forms of colorectal cancer. Mutations in at least one regulatory component of the Wnt pathway are present in over 93% of colorectal cancers [67]. Mutations in the *APC* gene are present in about 85%, whereas in 50% of tumors that do not contain the *APC* mutation, an activating β-catenin mutation is present [27,68]. The increased levels of nuclear β-catenin are associated with a poorer prognosis [69]. Similarly, mutations in the *RNF43* gene have been identified in up to 18% of colorectal cancers and are mutually exclusive with the *APC* mutations. *RNF43* encodes the E3 ubiquitin ligase needed for Frizzled receptor ubiquitination and inhibition of the Wnt pathway [11,67,70].

Another cancer in which Wnt signaling plays a major role is breast cancer. Studies have shown that Wnt signaling is active in over 50% of examined breast cancers subtypes, and this activity, higher in comparison to wild type cells, has been associated with poorer survival. It is most active in triple-negative breast cancers, where Wnt ligands and receptors are commonly overexpressed, whereas secreted Wnt antagonists are hypoexpressed. However, β-catenin itself is rarely mutated in breast carcinomas [27]. Deletions of Wnt pathway antagonist genes *DKK1*, *DKK3*, *WIF1*, *SFRP5*, *APC*, *GSK3B*, *MCC*, and *CTNNBIP1* were also found in 32–44% of tested samples and methylation of their promoters in 40–68%. miR-221/222 interferes with the transcription of Wnt pathway antagonists, including *WIF1*, *SFRP2*, *DKK2*, and *AXIN2*. It has also been shown that the Wnt pathway is most active in breast cancer stem cells relative to the rest of the tumor mass [71]. Preclinical studies have shown an overexpression of *FZD 6*, *7*, and *8* genes in triple-negative breast cancers, which has been associated with poorer survival, invasiveness, and tumor stem cell characteristics.

However, changes in the Wnt pathway need not be associated with a poorer prognosis in all tumors. Several studies report a less invasive phenotype and increased survival rate in melanoma patients with increased levels of nuclear β-catenin [27,72,73,74]. This is explained by the different influence on the Wnt pathway depending on the presence of different transcription factors [72]. Medulloblastoma is another type of tumor where Wnt activation is associated with a good prognosis. Mutations of the phosphorylation site of β-catenin are present in 18–22% of medulloblastomas, and *APC* or *AXIN1* mutations are present in another 5% of cases [75,76]. Of the four subgroups of medulloblastomas, the Wnt-positive subgroup has the best prognosis in terms of recurrence and metastasis and a five-year survival rate of 95% [76,77].

Studies investigating the role of Wnt signaling in hepatocellular carcinoma (HCC) have shown the activation of the pathway in about 25–50% of HCC as well as a tumor-specific mutational profile that differs to alterations found in CRC. In HCC, *AXIN1* or *CTNNB1* genes are frequently altered, whereas *APC* gene alterations are uncommon [67,78,79]. According to the cBioportal database of molecular changes in human cancers (https://www.cbioportal.org/, accessed on 7 March 2023), of 740 patients with hepatocellular carcinoma, 31% had a mutation in the β-catenin gene (https://bit.ly/3iXoIU5, accessed on 7 March 2023).

Several studies have shown that the excessive activation of the Wnt pathway is responsible for glioma formation and is associated with a poor prognosis [25,80,81,82]. In a study on astrocytomas of varying degrees of malignancy, the *SFRP1* gene was found to be hypermethylated in 32% of samples, and the SFRP1 protein expression was reduced in 45.8% of samples. *SFRP1* hypermethylation as well as enhanced LEF1 expression were associated with a higher tumor grade [83]. There is evidence that epigenetic changes play a more important role in Wnt pathway activation in glioblastomas than mutational changes. For example, the epigenetic attenuation of negative regulators *WIF1*, *SFRP1/2*, and *NKD1/2* (Naked) [84,85] by the hypermethylation of their promoters was found in about 40% of glioblastomas. Hypermethylation of the *DKK1* promoter was found in 60% of glioblastomas [86]. All these analyses suggest an important contribution of Wnt signaling to tumor formation and invasiveness [30,87,88]. In addition, the overexpression of Wnt genes has been associated with an increased resistance of glioblastoma cells to radiotherapy, whereas the inhibition of Wnt signaling by the small inhibitory molecules enhanced the sensitivity to radiotherapy [80]. Wnt inhibitors such as XAV939 also prevented glioma cell invasiveness and glial-mesenchymal transition [7].

The previously described research shows the important role of the Wnt signaling pathway in a large number of human tumors. Therefore, further research and development of inhibitors could prove a useful therapeutic strategy.

## 3. WNT Pathway Inhibitors

The reason for searching for inhibitors of the Wnt pathway is the hope that the inhibition would have a therapeutic effect on tumors [88,89]. For example, it has been observed that the silencing of β-catenin by siRNA has an inhibitory effect on the growth of colorectal cancers in vitro and in vivo. When knocking down β-catenin in colon cancer cell lines carrying the *APC* mutation, significant growth inhibition, differentiation, and reduction of proliferation occurred. However, after the cessation of beta-catenin silencing, tumor growth rapidly recovered, suggesting that Wnt inhibitor therapy would require continuous administration [90]. Another strategy for Wnt pathway downregulation is the inhibition of the β-catenin interaction with TCF, which has been shown to be antiproliferative and proapoptotic in adrenocortical tumor cell lines [91]. Moreover, compared to the knockdown of β-catenin, the knockdown of TCF4 has been shown to be more efficient [92]. Although the inhibition of the Wnt pathway by knocking down β-catenin has been successful in arresting tumor growth, it is accompanied by certain difficulties. β-catenin’s binding site for E-cadherin overlaps with sites for transcription factors. So, if our goal is to inhibit only the β-catenin localized in the nucleus, there is a problem with the size of the inhibitory molecule, which should be small enough to pass through the cell membrane and nuclear envelope and yet large and specific enough to be able to “cover” a relevant binding site of β-catenin [93]. A similar problem exists with the inhibition of the transcription factors and coactivators TCF, BCL9, and CBP. Molecules that meet these requirements have been found and will be described under the following subheadings.

Parallel studies have also focused on examining the effect of growth inhibition of other Wnt pathway components, for example, the *FZD7* gene. A reduced ability of tumor formation in mice was found after the transplantation of triple-negative breast cancer cell lines with knocked down *FZD7* [94]. A similar effect was shown in squamous cell carcinoma of the esophagus, where the lack of FZD7 inhibited cell growth, induced apoptosis, and suppressed migration [95]. Inhibition of Wnt-1 and Wnt-2 ligands by siRNA or specific antibodies has also been reported to promote apoptosis in non-small cell lung cancer cells [96].

Based on all these findings, molecular targets to which Wnt inhibitors are directed have been recognized, and the inhibitors are usually divided according to their site of action (Table 1).

In addition, inhibitors which enhance the activity of the so-called negative Wnt pathway regulators have also been developed. Most of these molecules have been considered as inhibitors based on screening, i.e., testing thousands of molecules, both synthetic and natural, to identify the most potent ones. Various tests were used to evaluate the inhibition, most commonly the TOPFlash assay, which shows how much the examined molecules interfere with β-catenin-dependent transcription of dTF12 (top flash-like luciferase reporter) [93]. In other cases, protein levels of β-catenin or other proteins were measured using immunohistochemical or Western blot analyses or on mRNA level using the qRT-PCR method. Such approaches have identified a very large number of molecules that inhibit the Wnt pathway. Inhibitors of the Wnt pathway are listed in the following subheadings according to the component on which they act. In order to summarize all the inhibitors and to give the reader the immediate location of the target sites, an illustration is given in Figure 2.

## 4. Porcupine Inhibitors

Porcupine (PORCN) is a member of the membrane-bound O-acyltransferases (MBOAT) that acylates (palmitoylates) Wnt ligands in the endoplasmic reticulum before their secretion into the intercellular space. Increased expression of this enzyme is associated with a poorer prognosis in squamous cell carcinomas of the head and neck [72]. PORCN-inhibiting molecules suppress the secretion of all Wnt ligands and are capable of inhibiting both canonical and non-canonical pathways [81]. The following oral PORCN inhibitors are known: WNT974 (LGK974), ETC-153 (ETC-1922159), RXC004, and CGX1321.

LGK974 (WNT974) (MedChemExpress, Brunswick, NJ, USA) is a small molecule able to decrease the expression of Wnt target genes. In renal carcinoma cell lines, LGK974 inhibits cellular proliferation and migration and increases the proportion of G1 cells [97,98]. In preclinical studies in a murine MMTV-Wnt1 mammary tumor model, growth delay was indicated by changes in tumor volume for the treated (T) and control (C) groups. The administration of LGK974 at doses of 1 or 3 mg/kg resulted in efficient tumor regression (T/C= −47% or −63%, respectively) on day 13 of treatment [99].

A similar effect of LGK974 was observed in a mouse model of the Wnt-dependent human head and neck squamous cell carcinoma (HNSCC) cell line in which a dose of 3.0 mg/kg caused substantial tumor regression of T/C% = −50% was achieved. In both studies, LGK974 did not cause significant weight loss in mice [99]. It has also been shown that this compound decreased epithelial ovarian cancer (EOC) cell viability in vitro and inhibited tumor growth in vivo. LGK974 is a hydrophobic molecule, poorly soluble in water, which is the reason for its poor bioavailability. However, complexed with cyclodextrin, it has an improved delivery. In a mouse lung cancer xenograft, LGK974 in complex with cyclodextrin showed less intestinal toxicity and a greater effect on tumor growth inhibition and survival [100]. LGK974 is currently in phase I clinical trials in patients with metastatic squamous cell carcinoma of the head and neck, patients with pancreatic cancer, triple-negative breast cancer, and cervical squamous cell carcinoma (NCT01351103; https://clinicaltrials.gov/ct2/show/NCT01351103, accessed on 19 February 2023). In patients with metastatic colorectal cancer containing WNT and BRAF mutations, LGK974 is on trial in combination with BRAF inhibitor LGX818 and cetuximab [8], and its side effects have been evaluated according to the Common Terminology Criteria for Adverse Events (CTCAE) for oncology drugs [101,102].

Another porcupine oral inhibitor with high bioavailability is ETC-153 (ETC-1922159) [103] (2023 Merck KGaA, Darmstadt, Germany). This small molecule inhibited growth in mouse models of breast cancer with Wnt1 overexpression by 52% and 78% at a daily dose of 1 and 3 mg/kg, respectively, without significant weight loss. It has also been demonstrated that fusion-bearing cancers will also be highly responsive to treatment with ETC-153. For example, in mouse xenografts, ETC-153 has been very successful in inhibiting the growth of human colon cancer with the fusion of the *R-spondin 2* and *3* genes. This indicates that R-spondin, which is a secreted agonist of the pathway, can safely and efficiently be inhibited with ETC-159 [103]. The tolerability of ETC-153 was evaluated in a dose-escalation phase I clinical trial in six cohorts of patients with advanced solid tumors by administering doses to a maximum of 30 mg. The side effects were observed in less than 20% of patients, indicating that ETC-159 was well tolerated [101].

Another small molecule inhibitor, CGX1321 (Curegenix Co. Ltd., Guangzhou, China), also specifically targets PORCN. It affects the production of Wnt ligands by inhibiting the post-translational palmitoylation and secretion of Wnt ligands in the endoplasmic reticulum. In such a manner, it can block downstream Wnt signaling. Thus, CGX1321 is effective in patients with an active canonical Wnt pathway. Currently, CGX1321 is being tested in clinical trials on colon cancer patients through oral administration [104].

## 5. WNT Ligand Antagonists

Suppression of Wnt signaling in the extracellular space is another potential territory for cancer therapy. Ipafricept (OMP54F28; IPA) (OncoMed Pharmaceuticals, Inc., Redwood City, CA, USA and Bayer, Leverkusen, Germany) is a recombinant fusion antibody consisting of an extracellular portion of FZD8 receptor and Fc fragment of human IgG1. The binding of Wnt ligands to Ipafricept prevents their interaction with FZD receptors [3,11,105]. In mouse patient-derived xenografts of pancreatic cancer, ipafricept showed a greater reduction in tumor growth compared to an older-generation chemotherapeutic, gemcitabine. The effect included a reduction in tumor stem cells as well as liver and lung metastases. All of these effects were, however, more pronounced in combination with gemcitabine [105]. In a phase I clinical study on 26 patients with solid tumors, patients were given different intravenous doses of ipafricept. For phase II clinical trials, a dose of 15 mg/kg every three weeks was recommended [3,5,11,105].

A phase Ib clinical trial for the combination of ipafricept with paclitaxel and carboplatin in recurrent ovarian cancers was also conducted. After the revision of the first protocol, doses per cohort were increased from 2 to 6 mg/kg resulting in the overall response rate of 75.7%, the median survival of 33 months, and the median progression-free survival of 10.3 months. Adverse reactions were observed in ≥15% of patients of which neutropenia was the most common. Bone toxicities at efficacy doses prevented further testing of this treatment regimen [106]. Another phase Ib dose-escalation clinical trial was initiated to evaluate the combination of ipafricept with nab-paclitaxel and gemcitabine in previously untreated patients with metastatic pancreatic adenocarcinoma. The study included 26 patients, who were given ipafricecept starting at 3.5 mg/kg in combination with standard doses of nab-paclitaxel and gemcitabine. The conclusion reached was that this combination can be administered with reasonable tolerance, but once again the study was terminated primarily due to bone-related toxicity [107]. The combination of ipafricept with sorafenib is a potential future approach for liver cancer and is in phase I clinical trial [108].

## 6. Frizzled Receptor Antagonists

Vantictumab (OMP-18R5) (OncoMed Pharmaceuticals, Inc., Redwood City, CA, USA and Bayer, Leverkusen, Germany) is a human IgG2 monoclonal antibody that inhibits FZD receptors, namely FZD1, FZD2, FZD5, FZD7, and FZD8. It binds to their extracellular domains and probably sterically inhibits the binding of Wnt ligands. Thereby, the phosphorylation of LRP6 co-receptors is blocked, β-catenin concentration is reduced, and Wnt signaling is stopped [109]. According to previous in vitro and in vivo studies, vantictumab inhibits tumor growth in multiple tumor types and reduces tumor regrowth. In mouse xenografts of breast and pancreatic tumors, vantictumab alone and in combination with taxanes or gemcitabine reduced the number of tumor-initiating cells [110,111,112,113]. Mouse xenografts of pancreatic adenocarcinoma and serous ovarian cancer showed a significant synergistic effect of vantictumab and ipafricept with nab-paclitaxel in the reduction of tumor size, whereas such synergism was not seen with gemcitabine. Similar findings were indicated in triple-negative breast cancer and non-small cell lung cancer xenografts. Additionally, the combination of vantictumab with paclitaxel was significantly more effective than paclitaxel alone in reducing tumor mass growth in HER2-negative breast cancer xenografts [114,115]. A phase Ib study enrolled patients with locally recurrent or metastatic HER2-negative breast cancer who were treated with weekly paclitaxel in combination with vantictumab. The combination was generally well tolerated with promising efficacy; however, the incidence of fractures limited future clinical development of this particular WNT inhibitor in metastatic breast cancer [113].

Vantictumab in monotherapy significantly slowed tumor growth in xenograft models of squamous cell carcinoma of the head and neck. Furthermore, a very potent effect of the combination of taxol and vantictumab on xenografts of breast and lung cancer was reported [110]. A further study aimed to engineer the synthetic antibody F2.A with specificity of FZD4 inhibition. F2.A inhibited tumor growth in ductal pancreatic adenocarcinoma cell lines [116].

A phase Ia clinical trial has been conducted for vantictumab on different solid tumors, including colon, breast, sarcoma, and neuroendocrine tumors. In most patients, vantictumab therapy was discontinued within 60 days due to disease progression, and in three elderly patients with neuroendocrine tumors, a prolonged phase of disease stabilization occurred [117]. However, several additional phase Ib trials have been conducted. In the first one, 31 previously untreated patients with metastatic pancreatic adenocarcinoma received increasing doses of vantictumab in combination with nab-paclitaxel and gemcitabine. After evaluating tumor mass reduction, 41.93% of patients showed a partial response to therapy, whereas in 12.9% of patients, the disease progressed [118]. In another, similarly structured trial on metastatic pancreatic cancer, half of the patients had a partial response to therapy and a third ended up with stable disease. Similar effects were observed in HER2-negative breast cancer [113].

Due to the important role of the Wnt pathway in bone metabolism, bone-related side effects and their frequency were one of the main concerns in all clinical trials. Vantictumab increased the concentration of the biomarker of bone degradation, βCTX, even at low doses; moreover, at doses greater than 5 mg/kg, it decreased the biomarkers of new bone formation, P1NP and osteocalcin [117]. The occurrence of bone side effects can be prevented by providing vitamin D, calcium supplements, and zoledronic acid. However, it has been shown that vantictumab will not produce the desired effect in tumors with mutations in one of the molecules descending from its site of action, the FZD receptor [118].

A chimeric monoclonal antibody OTSA101 (OncoTherapy Science, Inc. (OTS), Kawasaki City, Japan) is yet another FZD receptor antagonist directed specifically to FZD10. This antibody was developed after it was demonstrated that synovial sarcoma cells express significantly more FZD10 than normal tissues. Additionally, when it was radiolabeled with Yttrium-90, this radioactive form, OTSA101-DTPA-90Y, showed even stronger antitumor activity [3,5,119]. In mouse xenografts of synovial sarcoma with FZD10 overexpression, a substantial reduction in tumor mass occurred after a single administration of OTSA101-DTPA-90Y at a dose of 3.7 MBq. The median time to tumor progression in treated mice was 58 days compared with 9 days in the control group [119]. The recommended activity for further clinical investigations was 1110 MBq of 90Y-OTSA-101. However, grade 3 adverse reactions, most commonly hematological, occurred, so the authors recommend less energetic particle emitter radioisotopes such as Lutetium 177 as a better option [119].

## 7. LRP Co-Receptor Antagonists

Drugs that act antagonistically to the LRP co-receptors were originally registered for the treatment of parasite infections, but their potential for tumor treatment is currently being intensively investigated. Salinomycin (Abcam, Cambridge Biomedical Campus, Cambridge, UK) is a monocarboxylic polyether ionophore previously known for its antibiotic and coccidiostatic effects. Ionophores are small molecules that help specific ions cross the membrane. Numerous studies have shown that salinomycin also shows antitumor activity, especially against tumor stem cells. Although its exact mechanism of action has not been fully elucidated, it has been proposed that it is through the inactivation of the Wnt pathway. In addition to inducing LRP co-receptor degradation, salinomycin activates the transcription factor FOXO3a, which prevents β-catenin from binding to TCF [120,121]. Salinomycin reduces both phosphorylated and total LRP, leading to the inhibition of all downstream activity [122]. Besides Wnt, salinomycin has an inhibitory effect on other pathways: Akt, NF-kB, and Hedgehog. A study on glioblastoma cell lines showed that after treatment with salinomycin, negative regulation of cyclin D1 and Wnt1 proteins occurred, whereas [123] in chronic lymphocytic leukemia, salinomycin induced apoptosis [124]. Salinomycin has been established as a drug for targeting human cancer stem cells. It has also been shown that, when used with other therapies, it sensitizes chemodrugs or radiation [125].

Additional studies indicated a potential benefit from the combination of salinomycin with resveratrol, which led to the decreased expression of the EMT marker vimentin and the induction of apoptosis in breast cancer cells [126]. Although clinical trials have not yet begun, there is a legitimate concern that the toxicity of this drug could be significant, given the number of cellular pathways it affects as well as insufficient knowledge on the mechanisms of action.

Rottlerin (Abcam, Cambridge Biomedical Campus, Cambridge, UK) is a natural polyphenol isolated from the plant *Mallotus philippinensis*, which has historically been used as an antihelmintic drug. It affects a number of signaling pathways in tumors [127]. The effect of rottlerin on the Wnt pathway led to LRP6 receptor degradation [127]. In another study on adrenocortical cancer cell lines, Western blotting revealed reduced expression of LRP6 and β-catenin after rottlerin treatment [128].

Another antagonist to the LRP co-receptor is monensin (Abcam, Cambridge Biomedical Campus, Cambridge, UK), an antibiotic and antiparasitic ionophore derived from the bacterium *Streptomyces cinnamonensis* [129]. It showed antiproliferative, antimigrant, and pro-apoptotic effects in ovarian and pancreatic cancer, as well as synergism with oxaliplatin, erlotinib, and gemcitabine, respectively. Monensin has been shown to inhibit the Wnt signaling in intestinal tumors in vivo, without affecting healthy mucosal cells. It led to LRP6 degradation [129] and reduced cyclin D1 expression [130] and showed a selective cytotoxic effect on cells undergoing EMT [129].

Another oral antihelmintic drug that has recently been credited with effects on a wide range of diseases including tumors is niclosamide (Abcam, Cambridge Biomedical Campus, Cambridge, UK). The antiproliferative effect of niclosamide in colorectal, prostate, lung, ovary, and breast cancer by the inhibition of Wnt, mTOR, STAT3, Notch, and NF-kB signaling pathways has been reported [131]. It prevents the formation of spheroids in breast cancers and acts on the Wnt pathway by promoting LRP6 co-receptor degradation [132]. In addition, it promotes the degradation of FZD1 receptors DVL2 and β-catenin. The mechanism by which niclosamide degrades Wnt pathway participants is not fully understood, but it is probably mediated by autophagy because autophagy marker LC3 was found to colocalize with the pathway’s molecules after niclosamide administration [132,133,134]. However, cells with impaired autophagy or an efflux of niclosamide from the cell have been shown to be resistant to niclosamide [132]. In ovarian cancers, niclosamide reduced the number of tumor stem cells [135]. The nitro group in niclosamide has been associated with serious adverse events such as hepatotoxicity, mutagenicity, and bone marrow suppression, and the compound has poor bioavailability [135,136]. In order to increase its stability and prolong its half-life, analogs were produced: one in which the nitro group was replaced by a trifluoromethyl group and another in which the salicylic part of the molecule was modified. The analogs showed the same effects as niclosamide in ovarian and chemotherapy-resistant ovarian cancer cell lines, and their bioavailability was increased [135,137]. In a phase Ib clinical trial in prostate cancer patients, oral niclosamide was rejected for further trials because the maximum tolerated dose was not sufficient to cause a therapeutic effect [138].

## 8. Tankyrase Inhibitors

Tankyrase is an enzyme that regulates the stability of scaffolding proteins AXIN1 and AXIN2 by a reversible post-translational modification—poly-ADP-ribosylation. Such poly-ADP-ribosylated AXINs (PAR-ilylated AXINs) are recognized by ubiquitin ligase RNF146, which labels them for proteasome degradation. Due to the consequent decrease in the AXIN concentration, the formation of the β-catenin degradation complex is prevented [139]. Thus, AXIN limits the quantity of the β-catenin destruction complex. Several tankyrase-specific inhibitors for in vivo administration have been developed that act by lowering the concentration of beta-catenin. The first is the small-molecule inhibitor XAV939 (Abcam, Cambridge Biomedical Campus, Cambridge, UK; MedChemExpress, Brunswick, NJ, USA), which binds to the catalytic domain of tankyrase [140], leading to stabilization of AXIN and the β-catenin degradation complex [5,141]. The compound is a thiopyranopyrimidine, a member of (trifluoromethyl)benzenes. Compared to some other inhibitors, XAV939 is specific for Wnt signaling [140]. XAV939 showed an inhibitory effect on small cell lung cancer cell lines, alone and in combination with cisplatin [140], where a dose-dependent decrease in proliferation was observed [142]. A similar effect was achieved on HeLa cells, where XAV939 caused decreased viability and colony formation rate compared to radiotherapy alone. The combination of XAV939 with radiotherapy led to a decrease in the expression of Wnt proteins Wnt3a, Wnt5b, β-catenin, cyclin D1, and c-myc and increased the ratio of apoptotic cells by 46.53% compared to radiotherapy alone [143,144]. Another study on lung adenocarcinoma A549 cells showed reduced expression of tankyrase, β-catenin, and c-Myc protein in response to XAV939 administration as well as the decreased viability, proliferation, and migration relative to the control group [145]. Furthermore, the combination of paclitaxel and XAV939 was effective in reducing the viability of triple-negative and ER+ breast cancer and significantly reduced tumor growth in mouse xenografts compared to monotherapies (at doses of 10 mg/kg) [146].

Further tankyrase inhibitors include JW-55, JW-74, and G007-LK (Sigma-Aldrich Pty Ltd., An affiliate of Merck KGaA, Darmstadt, Germany; MedChemExpress, Brunswick, NJ, USA). Generally, they all increase cytoplasmic AXIN levels, β-catenin degradation, and downregulate the expression of Wnt target genes. In colorectal cancer cell lines with *APC* mutations, the application of JW-74 resulted in the reduction of tumor growth and cell cycle arrest in the G1/S phase [5,144]. Tankyrase inhibitor G007-LK reduced the expression of Wnt and Hippo signaling pathway proteins [147]. In glioma, it showed antiproliferative activity, reduced glioma stem cell sphere formation, and potentiated the effect of temozolomide. Another study showed that orally administered doses of G007-LK reduced the proliferation frequency of the LGR5+ intestinal stem cells in mice without affecting tissue morphology [148]. The LGR5+ stem cells are located in the crypt base and are capable of regenerating all intestinal epithelial cell lineages. This inhibitor was well tolerated in mice, without disrupting the structure and function of the intestine.

Another tankyrase inhibitor is the spirindoline derivative RK-287107 (Abmole Bioscience Inc., Houston, TX, USA; MedKoo Biosciences, Inc., Morrisville, NC, USA) [149]. When administered orally at tolerable doses, it inhibited the growth of colorectal tumors in vitro and in vivo. This drug is considered to be the most likely candidate from the group of tankyrase inhibitors for future clinical trials. RK-287107 inhibits both tankyrase 1 and 2 expression [139] and has been shown to be highly selective for tankyrase (PARP5 a and b) over other PARP enzymes (PARP 1, 2s and 10) [149]. RK-287107 inhibited the growth of the colorectal cancer cell line COLO-320DM, which harbors a short form of *APC*. The concentration which reached 50% growth inhibition was 0.449 μmol/L. However, it showed no significant effect on the growth of cell lines with gain-of-function *CTNNB1* mutations, lines with longer forms of APC mutant proteins, or lines with wild-type APC [139]. Of note is that the status of the *APC* mutation is an important factor for sensitivity to tankyrase inhibitors in colorectal cancer. Immunohistochemistry showed that both RK-287107 and G007-LK inhibitors reduced unphosphorylated active β-catenin levels. RK-287107 also induced the accumulation of AXIN2 and the downregulation of MYC. The next experiment aimed to determine whether RK-287107 is available for oral dosage. Intraperitoneal (150 mg/kg twice daily) and oral (300 mg/kg twice daily) administration of this drug in mouse xenografts with *APC* mutations that produce shorter protein forms resulted in 47.2% and 51.9% tumor growth inhibition, respectively. Pharmacokinetic analysis revealed higher plasma concentrations after oral than after intraperitoneal administration [139], and its bioavailability was about 60% [5,149].

Currently, there are two other tankyrase inhibitors in preclinical phases: LZZ-02 (Medkoo Biosciences, Morrisville, NC, USA) and NVP-TNKS656 (Merck KGaA, Darmstadt, Germany). A combination of NVP-TNKS656 and PI3K and AKT inhibitors has shown to be a promising tool for decreased beta-catenin translocation to the nucleus and Wnt silencing in PI3K or AKT inhibitor-resistant cells of colorectal carcinoma. A similar result on the restraint of active β-catenin was noted when the novel tankyrase inhibitor LZZ-02 was used.

## 9. Molecules That Promote Proteasomal Degradation of β-Catenin

One of the novel strategies for the inhibition of the aberrant Wnt signaling pathway is through the restoration and stabilization of the activity of the molecules responsible for the degradation of β-catenin. One such molecule is pyrvinium (Merck KGaA, Darmstadt, Germany), an anthelmintic drug already approved by the FDA that exerts its antitumor effects by restoring the activity of several different kinases that are directly or indirectly involved in β-catenin degradation. It is important to highlight that pyrvinium pamoate is approved as antihelmintic drug, but at present the FDA approval does not extend to its antitumor effect. A recent study showed that pyrvinium directly binds to CK1α as an activator, stabilizing its protein [150,151,152]. The first step leading to β-catenin degradation involves the CK1α-initiated phosphorylation at Ser45 of β-catenin, followed by glycogen synthase kinase 3β (GSK3β) phosphorylation at Ser33, Ser37, and Thr41 to promote β-catenin proteasomal destruction [153]. Therefore, the pyrvinium-mediated restoration of CK1α activity inhibits Wnt signaling. Pyrvinium can also act inhibitory on AKT kinase, an indirect negative regulator of GSK3β [154]. If GSK3β is deactivated, β-catenin’s phosphorylation and degradation is also prevented [155]. Thus, by acting inhibitory on AKT kinase, pyrvinium will also restrain Wnt signaling. Recent findings suggest that pyrvinium selectively potentiates CK1α kinase activity [156,157]. Another avenue for pyrvinium action is by PYGO96 degradation that will once again lead to the negative regulation of the transcriptional activity of β-catenin [5]. Colon cancer cell lines with *APC* mutations were sensitive to pyrvinium treatment, showing a decrease in cell proliferation. Furthermore, pyrvinium targets ovarian cancer cells through suppressing Wnt signaling [158].

A new potent and selective inhibitor of Wnt signaling that acts by promoting β-catenin proteosomal degradation is MSAB (Sigma-Aldrich Pty Ltd., An affiliate of Merck KGaA, Darmstadt, Germany), or methyl-3 (4-methylphenyl) sulfonyl amino-benzoate. This was recognized by the screening of 22,000 molecules in a colorectal cancer cell line harboring a deletion of β-catenin’s phosphorylation site of CK-1 [159]. It has been shown that MSAB binds to β-catenin [5] and reduces its nuclear levels while slightly increasing its cytoplasmic levels. MSAB has also been shown to increase the fraction of phosphorylated β-catenin [160]. The degradation of β-catenin happens most likely through the binding of MSAB to β-catenin’s armadillo domain [45].

The selectivity of MSAB for tumor cells was demonstrated in epithelial cells or skin fibroblasts. MSAB also caused a reduction in tumor mass in mouse xenografts in Wnt-dependent tumors, causing apoptosis of tumor cells. It is important to note that MSAB also inhibited the proliferation of LS174T human colon adenocarcinoma cell line carrying β-catenin mutations [159]. The future development of β-catenin inhibitors should be aimed at finding molecules that would be selective for a particular type of mutational variant of β-catenin and thus spare non-tumor tissues [43,93,159].

## 10. β-Catenin and TCF Complex Inhibitors

The next section will focus on the inhibitors that target the downstream protein–protein interactions of β-catenin with transcription factors and coactivators. Such inhibition is primarily by blocking the β-catenin/TCF complex. Several compounds, such as PKF115-584, CGP049090 (MedKoo Biosciences, Inc., Morrisville, NC, USA), and PKF222-815 (Novartis Pharmaceuticals, Inc., Basel, Switzerland), acting in a dose-dependent manner were identified using high-throughput ELISA screening. However, these molecules proved to be insufficiently selective because they also inhibited the APC/β-catenin interaction [43]. Nevertheless, in tumors with mutated *APC*, these compounds could prove very useful [161].

In a study performed by Huang et al. [161], a selective binding site that can differentiate β-catenin/TCF, β-catenin/cadherin, and β-catenin/APC interactions was identified through the investigation of selective small-molecule inhibitors. Potent inhibitors were discovered that completely disrupt β-catenin/TCF interactions. The small molecules UU-T02 and UU-T03 (MedChemExpress, Brunswick, NJ, USA) were found to be the most selective in inhibiting interactions of β-catenin and TCF. UU-T03 effectively reduced the expression of Wnt target genes and the growth of colorectal cancer in vitro, with selectivity for tumor over normal cells. It should be pointed out that hydrophobic B and C pockets were recently identified on the β-catenin. They are specific for TCF interaction and therefore represent a potential avenue for future development of more selective drugs [43,161].

## 11. β-Catenin and CBP Complex Inhibitors

In addition to TCF, the coactivators LEF, CBP, p300, and PYGO also form complexes with β-catenin. The co-activator CREB binding protein (CBP) is a particularly interesting target. Several CBP inhibitors have been developed in recent years, including PRI-724 (Prism Pharma Co Ltd. (Prism Pharma), Yokohama, Japan), a small molecule that inhibits the interaction of β-catenin with CBP by competitively binding to CBP. Preclinical studies have shown that PRI-724 displays antitumor effects, promotes the differentiation of tumor stem cells, and makes them more sensitive to chemotherapy [69,101,160]. In a phase I clinical trial, PRI-724 was tolerated in patients with solid tumors at a dose of 905 mg/m^2^ in continuous infusion for 7 days, and the inhibitor proceeded to phase II trial [162]. In a phase Ib clinical trial on metastatic pancreatic cancer, PRI-724 was given in combination with gemcitabine after various 5-FU-based regimens, including FOLFOX or FOLFIRINOX. The patients were divided into three cohorts receiving PRI-724 at escalating doses of 320, 640, and 905 mg/m^2^/day. The response was the stabilization of disease in 40% of patients. Grade 3 and 4 adverse events were observed in seven patients, but none of the side effects were dose-limiting [101,163].

The peptide CWP232291 (JW Pharmaceutical, Seoul, Republic of Korea) is another potent inhibitor of β-catenin and CBP complex formation [101]. Two phase Ia trials were conducted: one on refractory multiple myeloma and the other on acute myeloid leukemia and myelodysplastic syndrome. Multiple myeloma patients received intravenous doses of 198–446 mg/m^2^. Two dose-limiting toxicities occurred at the highest dose: grade 3 and 4, hypoxia and thrombocytopenia. The continuation of phase Ib is planned to evaluate the combination therapy of CWP232291 with lenalidomide and dexamethasone [164]. In a second clinical trial on acute myeloid leukemia and myelodysplastic syndrome, patients received CWP232291, which resulted in complete remission in one patient at doses of 153 and 118 mg/m^2^ [101,165].

## 12. β-Catenin and BCL9 Complex Inhibitors

BCL9 (B-cell lymphoma 9) is another coactivator for β-catenin-mediated transcription that is highly expressed in tumors. Carnosic acid represents a pharmacologic strategy for inhibiting oncogenic BCL9 and β-catenin interaction by blocking the binding site for BCL9 on β-catenin [5,166]. Carnosic acid (Merck KGaA, Darmstadt, Germany) is a weak acid. In terms of its chemical structure, it is a phenolic diterpene, a natural compound isolated from rosemary, with antioxidant and antimicrobial properties. Besides carnosic acid, other molecules that inhibit the BCL9 and β-catenin complex are sulfono-γ-AApeptides (CSBio, Menlo Park, CA, USA). They mimic the structure of the BCL9 α-helix, which allows them to bind to β-catenin and impair its binding to BCL9. In addition, sulfono-γ-AApeptides have shown marked stability and resistance to proteolysis, a promising therapeutic characteristic [43,167,168].

## 13. Inhibitors of CLK Kinases

CDC-like kinases (CLKs) are evolutionary conserved kinases that are able to phosphorylate serine, threonine, and tyrosine residues. CLKs have primarily been involved in precursor-mRNA splicing, where they catalyze the phosphorylation of splicing factors 1–12 (SRSF1-12). In spite of the fact that the knowledge about their biological roles is still rudimentary, the therapeutic potential of CLK inhibitors has been extensively explored, and the compound SM08502 (Cirtuvivint) (Biosplice Therapeutics, Inc., San Diego, CA, USA) has recently entered clinical trials [169]. It is a small molecule that can be administered orally. In an inhibition trial tested on 402 kinases, SM08502 singled out kinases CLK2, CLK3, CLK1, CLK4, and DYRK [169]. In colorectal carcinoma cells, SM08502-mediated the reduction of Wnt pathway activity through a strong inhibition of splicing factor SRSF phosphorylation and the disruption of alternative splicing. Later, it was shown that SM08502 reduced the level of phosphorylated splicing factors SRSF 5 and 6 [168], which led to intron retention in *VL2*, *TCF7*, *ERBB2*, and *LRP5* genes and exon skipping in *LEF1* and *TCF7L2* genes, making the resulting mRNA unstable. SM08502 was 10 times more potent than PRI-724, the inhibitor of β-catenin transcriptional activity. Moreover, SM08502 inhibited spheroid formation in pancreatic cancer cell lines. SM08502 induced apoptosis and a significant reduction in tumor growth in all gastrointestinal tumor cell lines in vitro and in mouse colorectal cancer xenografts [169]. The effect of oral SM08502 on tumor growth was evaluated in CRC and gastric cancer xenograft models in athymic nude mice. The maximum tolerated dose in mice was 50 mg/kg/day [169]. Similar findings were reported for triple-negative breast and ovarian cancers [170]. SM08502 inhibited the proliferation of breast and prostate cancer cell lines. Therapeutic effects improved with the combination of SM08502 and docetaxel. In patient-derived xenograft and cell line-derived xenograft models of pancreatic cancer, SM08502 in combination with gemcitabine and paclitaxel caused significant tumor regression compared to gemcitabine and paclitaxel alone [171]. Because preclinical results in vivo and in vitro have been promising, SM08502 is currently in phase I clinical trials.

Other compounds have been employed to targeted CLKs. TG003 (MedChemExpress, Brunswick, NJ, USA) has shown potent antitumor properties in prostate cancer cells in vitro and in a xenograf model and caused significant changes in the alternative splicing of cancer-associated genes. The use of compounds CC-671 (Cayman Chemical, Ann Arbor, MI, USA) and T-025 (MedChemExpress, Brunswick, NJ, USA) also resulted in significant antitumor effects [172].

## 14. Side Effects of WNT Inhibitors

Because an optimally regulated Wnt pathway ensures cellular differentiation as well as the regeneration of many tissues, especially those with rapid cell turnover such as hematopoietic and gastrointestinal tissues, the inhibition of this pathway carries serious risks of side effects. Side effects can be expected when inhibiting such an essential cellular pathway. Nevertheless, early clinical trials have shown that the severity of side effects is not significant compared to the therapeutic benefits. They are usually associated with the malfunction of tissue regeneration [11] and could drastically limit the use of Wnt pathway inhibitors in systemic antitumor therapy.

The most common side effects upon inhibition of Wnt signaling are gastrointestinal problems, hair loss, immunosuppression, fatigue, vitiligo, anemia, neutropenia, thrombocytopenia, bone fractures, and neurodegeneration. In addition to these adverse reactions, elevations in bilirubin and alkaline phosphatase have been observed, as well as hypophosphatemia. This could be explained by the change in bone remodeling, but also by the fact that proper Wnt signaling is important in the regeneration of liver and kidney tissues after injury [173]. However, the weak point of many studies is the lack of data on prior injuries or kidney diseases.

Because the Wnt pathway regulates bone remodeling in a complex way, one of the side effects reported in the early clinical stages is an increase in bone remodeling. However, it has been shown that this can be prevented with bisphosphonates, specifically zolendronic acid. It is known that decreased LRP5 expression in mouse osteoblasts leads to an osteopenic phenotype, whereas increased LRP5 expression results in increased bone mass [174]. The activation of the Wnt pathway promotes the differentiation of mesenchymal progenitor cells into osteoblasts [3,175], and β-catenin stimulates osteoprotegerin expression in differentiated osteoblasts, which by binding to the protein RANKL contribute to the inhibition of osteoclast differentiation. Thus, the activation of the Wnt pathway shifts the balance toward bone synthesis [176]. Pathological fractures during the pharmacological inhibition of the Wnt pathway could be explained by these observations [174].

However, novel research indicated [177] that a clinically approved anti-resorptive, alendronate, could mitigate the loss of bone mass and extend the beneficial antitumor effects of PORCN inhibitors.

Needless to say, all Wnt pathway inhibitors are contraindicated in pregnancy. Wnt signaling is extremely important in development. For example, Wnt signaling gradients control the establishment of the anterior–posterior axis in the central nervous system and also play a role in dendritic and axon guidance and synaptogenesis [7,178].

In all phases of clinical trials, the question remains as to how the inhibition of the Wnt pathway would affect the cognitive abilities of patients, especially the younger ones. In the adult brain, the proper activation of the Wnt pathway is crucial for adult neurogenesis and the survival of neurons in the supraventricular zone and hippocampus, but also for the maintenance of higher cognitive functions and dopaminergic pathways, and it appears to affect synapse formation [1,178]. Murine studies are not optimal models for such side effects.

All in all, the side effects caused by Wnt inhibitors are similar to those of chemotherapeutics that are already known and well established. However, it is important to distinguish whether the reported side effects are strictly due to Wnt inhibitors or whether they are the consequence of chemotherapeutics in combinational therapy. Interactions with other molecules which can lead to the so-called off-target side effects have also been insufficiently investigated [43]. To overcome the side effects of systemic Wnt inhibition, strategies have been proposed for delivering the inhibitors directly to tumor cells by using nanoparticles, liposomes, or binding the inhibitor to one of the molecules attracted by a particular tumor [72].

## 15. Combination of WNT Inhibitors with Other Forms of Antitumor Therapy

The beneficial effect of Wnt inhibitors on other modalities of antitumor therapy was also investigated. The first interesting possibility was alleviating the resistance to checkpoint inhibitors by using Wnt inhibitors [179]. It has been shown that the upregulation of Wnt signaling can cause resistance to immune checkpoint inhibitor therapy by modulating the tumor microenvironment through the interaction with tumor-associated macrophages, or by stimulating an acidic tumor environment that is immunosuppressive to cytotoxic T lymphocytes. Thus, Wnt signaling helps the so-called immune cell exclusion, preventing immune cells from reaching the tumor, and the tumor becomes resistant to checkpoint inhibitor therapy [180]. Several preclinical studies have shown that inhibition of the canonical Wnt pathway in parallel with the use of checkpoint inhibitors can effectively overcome this resistance [180].

In line with these findings are the results of a phase 1 clinical trial using the porcupine inhibitor LGK974 in combination with spartalizumab, a monoclonal antibody to PD-1, that reported impressive results in patients with several types of solid tumors, including the stabilization of disease in 53% of urothelial carcinoma previously resistant to checkpoint inhibitors [180].

Besides influencing immunotherapy, the combination of Wnt inhibitors with taxanes also demonstrated superior clinical response. Taxanes, including paclitaxel, nab-paclitaxel, and docetaxel, block the M-phase of cell division by acting on microtubules. Wnt pathway components are involved in the cell cycle, too. β-catenin is necessary for the separation of centrosomes in the formation of the mitotic spindle, whereas APC and Dishevelled participate in the regulation of kinetochores binding and, together with FZD and LRP, affect the orientation of the spindle [181]. Thus, the inhibition of Wnt signaling leads to spindle defects. Nab-paclitaxel in mice xenografts of pancreatic cancer caused an increase in the number of cells in the G2-M phase and a three-fold increase in β-catenin levels in mitotic cells. The synergism of Wnt inhibitors with taxanes could be generally explained by a dual effect on the disruption of cell divisions and also the prevention of the Wnt pathway activation after taxane administration. The combination of ipafricept and vantictumab with taxanes was evaluated on mouse xenografts of breast, ovarian, and pancreatic tumors. Wnt inhibitors have been shown to potentiate the cytotoxic effect of taxanes by modulating Wnt pathway activity in mitotic cells but are less effective in combination with S-phase blockers or platinum-based drugs. The optimized protocol of vantictumab and ipafricept, 25 mg/kg every 2 or 3 weeks, proved to be more effective and less toxic to the bone. For the optimal effect of this combination, it is necessary to administer the Wnt inhibitor before taxane because reverse or concomitant use has been shown to be less effective [11,106,110,112,154,182]. Clinical trials of this drug combination have been described in the section on ipafricept.

## 16. Conclusions

Numerous studies have yielded molecules that are capable of inhibiting abnormal Wnt signaling activity in tumor cells. The preclinical and clinical studies for many of these molecules show promising results. However, the mutational status of Wnt components has not been adequately addressed, so the question remains whether a particular inhibitor would have an effect on their mutated forms. The majority of reported studies did not monitor disease progression or recurrences, but instead only measured the inhibitory effect on tumor size or mass. For future investigations, it would be necessary to distinguish the specific tumor in which a particular Wnt inhibitor is successful. The most likely candidate for future antitumor therapies is the PRI-724 inhibitor, which is currently in phase II clinical trials. Given the many serious side effects caused by Wnt inhibitors, as well as the insufficient knowledge of the additional adverse effects, further research is recommended on the safety, efficacy, and drug delivery modes. Drugging the Wnt signaling pathway continues to be one of the promising approaches for future tumor treatment, both alone and in combination therapy.

## Figures and Tables

**Figure 1 ijms-24-06733-f001:**
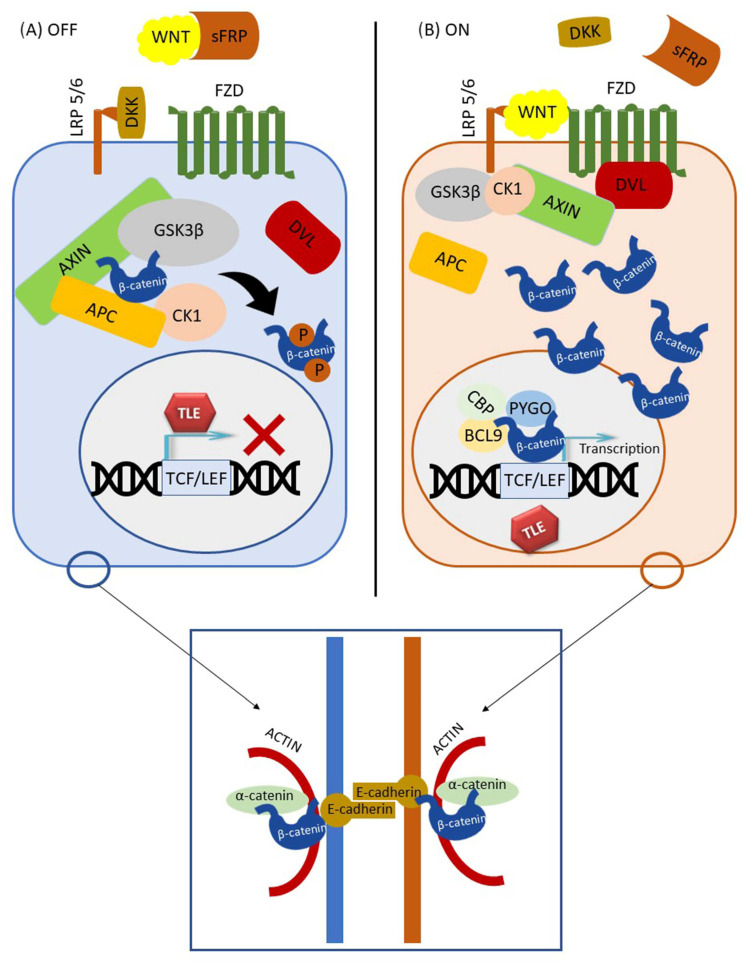
Canonical Wnt signaling pathway. Two modes of signaling are shown: (**A**) inactive and (**B**) active. Adherens junction is shown in the blue square.

**Figure 2 ijms-24-06733-f002:**
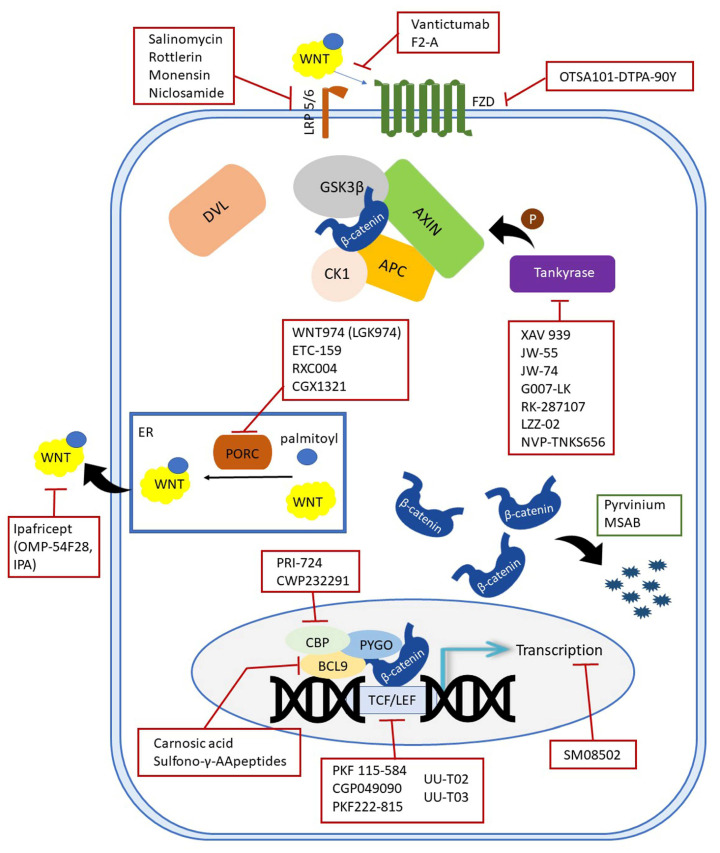
Schematic representation of inhibitors targeting the canonical Wnt components.

**Table 1 ijms-24-06733-t001:** Inhibitors of the canonical Wnt pathway. Adapted with permission from Ref. [5]. Courtesy of Professor Park.

Wnt Molecules Inhibitors	Compound	Cancer Type	Phase/Identification Number
PORCN Inhibitors	WNT974 (LGK974)	Squamous cell carcinomas of the head and neck	phase 2 NCT02649530
		Pancreatic cancer Colorectal cancer with BRAF mutation Melanoma Triple-negative breast cancer Squamous cell carcinomas (head and neck, cervix, esophagus, lungs)	phase 1 NCT01351103
		Metastatic colorectal carcinoma (with LGX818 and cetuximab)	phase 1 NCT02278133
	ETC-153 (ETC-1922159)	Solid tumors	phase 1 NCT02521844
	RXC004	Solid tumors	phase 1 NCT03447470
	CGX1321	Colorectal cancer	phase 2 NCT04907539
		Colorectal adenocarcinoma Gastric adenocarcinoma Pancreatic adenocarcinoma Bile duct cancer Hepatocellular carcinoma, Esophageal cancer Gastrointestinal cancer	phase 1 NCT03507998
		Solid tumors Gastrointestinal cancers (with pembrolizumab)	phase 1 NCT02675946
WNT ligand antagonist—an inactive FZD8 decoy receptor	Ipafricept (OMP-54F28, IPA)	Solid tumors	phase 1 NCT01608867
		Ovarian cancer (with paclitaxel and carboplatin)	phase 1 NCT02092363
		Metastatic pancreatic carcinoma (with nab-paclitaxel and gemcitabine)	phase 1 NCT02050178
		Hepatocellular carcinoma (with sorafenib)	phase 1 NCT02069145
Frizzled receptor antagonists	Vantictumab (OMP-18R5)	Solid tumors	phase 1 NCT01345201
		Metastatic breast cancer (with paclitaxel)	phase 1 NCT01973309
		Metastatic pancreatic carcinoma (with nab-paclitaxel and gemcitabine)	phase 1 NCT02005315
		Solid tumors (with docetaxel)	phase 1 NCT01957007
FZD10 antagonist	OTSA101-DTPA-90Y	Synovial sarcoma	phase 1 NCT01469975
Synthetic antibody against FZD4	F2.A	Preclinical	
Tankyrase inhibitors	XAV939 XAV939 with cisplatin XAV939 with paclitaxel	Preclinical	
	JW-55 and JW-74 G007-LK RK-287107 LZZ-02 NVP-TNKS656	Preclinical	
CBP/β-catenin antagonists	PRI-724	Advanced pancreatic cancer Metastatic pancreatic cancer Pancreatic adenocarcinoma	phase 1 NCT01764477
		Advanced solid tumors	phase 1 NCT01302405
		Acute myeloid leukemia Chronic myeloid leukemia	phase 2 NCT01606579
		Acute myeloid leukemia Chronic myeloid leukemia (with leucovorin calcium, oxaliplatin or florouracil)	phase 2 NCT02413853
	CWP232291	Multiple myeloma Acute myeloid leukemia Myelodysplastic syndrome	Phase 1 NCT01398462 NCT02426723
Inhibitors of β-catenin-controlled gene expression	SM08502	Solid tumors	phase 1 NCT03355066
β-catenin/TCF complex inhibitors	PKF115-584 CGP049090 PKF222-815	Preclinical	
	UU-T02 UU-T03	Preclinical	
β-catenin and BCL9 complex inhibitors	Carnosic acid Sulfono-γ-AApeptides	Preclinical	
LRP coreceptor antagonists	Salinomycin	Preclinical	
	Rottlerin	Preclinical	
	Monensin	Preclinical	
	Niclosamide	Colon cancer	Phase 1 (terminated) NCT02687009
		Metastatic Prostate Carcinoma	Phase 1 NCT03123978
Molecules that promote proteasomal degradation of β-catenin	Pyrvinium	Pancreatic cancer	Phase 1 NCT05055323
	MSAB	Preclinical	

## Data Availability

Data is contained within the article.

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
