# Peer review of "Wnt Signaling Inhibitors and Their Promising Role in Tumor Treatment"

_ijms, 2023, doi:10.3390/ijms24076733_

Round 1
Reviewer 1 Report
This is a well conceptualized and written review on the plethora of the Wnt pathway inhibitors investigated in preclinical and clinical settings. Authors paid a great deal of attention to defining the pathway before discussing various novel investigational agents. Table 1. nicely summarizes discussed agents, and provided clinicaltrials.gov identifier numbers are super helpful for readers who would like to do additional research on individual trials. In my mind, this table should appear at the start of the chapter discussing Wnt inhibitors. Also, summarizing the side-effects gives a needed balance to the described efficacy in previous chapters.
The main criticism that needs to be addressed before publishing is the lack of references in Figures 1 and 2, as well as in the Table 1. Namely, it is unclear if both Figures and the Table are copied (with permissions) or adapted from original reference(s). At any rate, sources need to be clearly identified and referenced.
Another important missing information in the review are manufacturers of discussed investigational agents. The manufacturer name should be added in the brackets whenever a new compound is mentioned for the first time, so that readers may either gain more information on corporate web sites, or know where to seeks a compound for own research, if so desired.
Correcting the following minor comments will also increase the quality of the review:
Line 142 - "metastasis" needs to be replaced with "metastatic spread";
Line 162 - "the next cancer" should probably read "another cancer";
Line 205 - "a desirable effect" is probably "a therapeutic effect";
Line 463 - Please clarify that pyrvinium pamoate is the FDA approved anthelmintic drug, and that the FDA approval does not extend to antitumor effects.
Author Response
Dear Editor,
enclosed please find our revised review article titled: "Wnt signaling inhibitors and their promising role in tumor treatment", manuscript ijms-2313741, which may be accepted pending minor revisions. We corrected the manuscript according to reviewers’ suggestions and answered their questions. We would like to thank you for all the suggestions and comments for improving our manuscript. Let me list all the revisions and responses
Comments and Suggestions from Reviewer 1.
1.Table 1 should appear at the start of the chapter discussing Wnt inhibitors.
We moved the Table 1 as suggested by the Reviewer.
2.The main criticism that needs to be addressed before publishing is the lack of references in Figures 1 and 2, as well as in the Table 1. Namely, it is unclear if both Figures and the Table are copied (with permissions) or adapted from original reference(s). At any rate, sources need to be clearly identified and referenced.
The table and figures were not copied from other articles. The figures were drawn by co-autor A.B. However, Figure 2 was adapted from Ahmed et al. (2016), van Andel et al. (2019), and Jung and Park (2020), which we now indicated in the corrected article on page 13, lines 248-249: Adapted from [5,137,176]. We also added reference in Table 1. since it was also partially adapted from Jung and Park, 2020 [5].
- Another important missing information in the review are manufacturers of discussed investigational agents. The manufacturer name should be added in the brackets whenever a new compound is mentioned for the first time, so that readers may either gain more information on corporate web sites, or know where to seeks a compound for own research, if so desired.
We added manufacturers' names in brackets at first mention of each compound.
- Minor comments were corrected:
Line 142 - "metastasis" was replaced with "metastatic spread";
Line 162 - "the next cancer" with "another cancer";
Line 205 - "a desirable effect" with "a therapeutic effect";
Line 463 - Please clarify that pyrvinium pamoate is the FDA approved anthelmintic drug, and that the FDA approval does not extend to antitumor effects.
We corrected the text to avoid confusion. It reads as follow on page 22, lines 489,490 :
Important to highlight is that pyrvinium pamoate is approved as antihelmintic drug, but at present the FDA aproval does not extend to antitumor effect.
All the corrections in the text are in track changes and markups. We believe that the revised article would be interesting to the readers and could be accepted for publication in your esteemed journal. We would like to thank you and the editors for your valuable suggestions and help.
Yours sincerely,
Nives Pećina-Šlaus

Reviewer 2 Report
Nicely written comprehensive Review on the Wnt pathway with emphasis on Wnt-inhibitors. I like the overview on the inhibitors in Fig. 2. Also Table 1 is very helpful.
Very helpful is the long reference list.
This is a very detailed overview on Wnt-inhibitors applied in the clinic aimed at various different kind of cancers make this review a compendium for clinical research.
Important point:
I miss one sentence in the introduction on the first discovery of the Wnt gene and its most important members.
Minor points
51: cannonical
53-54: “There are 19 different Wnt ligands.” Which species is meant?
102: NLS refers to the nuclear localization signal; so you can speak of nuclear localization signal sequences
136: the two domains should be included in this sentence
150: APC and CTNNB1 are abbreviations. Please outline where these come from. See also RNF (158). In general: please give outlines for all abbreviations used perhaps in an additional text-box
162: omit certainly
162-164: is Wnt activity in breast cancer too high in comparison to WT cells where Wnt activity is also present?
236: inhibitors of the Wnt pathway
Throughout the Review: the text could be better structured in that e. g. the different cancer forms, drugs or the different inhibitors mentioned are written in bold letters
415: What kind of small molecule is XAV939, as this was done for many other small molecule inhibitors
425: relative to a / the control group
500: potent inhibitors, not a inhibitors
612: aslo
Author Response
Dear Reviewer,
enclosed please find our revised review article titled: "Wnt signaling inhibitors and their promising role in tumor treatment", manuscript ijms-2313741, which may be accepted pending minor revisions. We corrected the manuscript according to reviewers’ suggestions and answered their questions. We would like to thank you for all the suggestions and comments for improving our manuscript. Let me list all the revisions and responses
Comments and Suggestions from Reviewer 2.
- I miss one sentence in the introduction on the first discovery of the Wnt gene and its most important members.
We added the sentence in the introduction on the first discovery of the Wnt gene and its most important members onpage 2 , lines 43-47 as follows:
The Wnt signaling is a conserved cellular pathway in all multicellular organisms that has been studied for more than four decades. The name was coined from the names of two genes, mouse int-1 and Drosophila's wingless (wg). The discovery of a novel cellular proto-oncogene int-1 which was later on mapped to the chromosomal position of Drosophila gene wg, launched the marvelous research on this essential pathway and its many important components.
2.Minor points were corrected:
51: cannonical
53-54: “There are 19 different Wnt ligands.” Which species is meant? Mammals and humans
102: NLS refers to the nuclear localization signal; was corrected to nuclear localization signal sequences
136: the two domains were included in this sentence
150: APC and CTNNB1 are abbreviations. Please outline where these come from. See also RNF (158). In general: please give outlines for all abbreviations used perhaps in an additional text-box
We included in the article the additional abbreviations in text-box.
162: word certainly was omitted
162-164: is Wnt activity in breast cancer too high in comparison to WT cells where Wnt activity is also present?
We corrected the sentence as follows lines 172, 173: Studies have shown that Wnt signaling is active in over 50% of examined breast cancers subtypes, and this acitvity, higher in comparison to wild type cells, has been associated with poorer survival.
236: inhibitors of the Wnt pathway
Throughout the Review: the text could be better structured in that e. g. the different cancer forms, drugs or the different inhibitors mentioned are written in bold letters.
We checked with IJMS style and formatting and it is not usual to bold all the cancer forms, drugs etc.
415: What kind of small molecule is XAV939, as this was done for many other small molecule inhibitors
The compound is a thiopyranopyrimidine, a member of (trifluoromethyl)benzenes. We added the explanation in the text.
425: relative to a / the control group
500: potent inhibitors, not a inhibitors corrected
612: aslo corrected
All the corrections in the text are in track changes and markups. We believe that the revised article would be interesting to the readers and could be accepted for publication in your esteemed journal. We would like to thank you and the editors for your valuable suggestions and help.
Yours sincerely,
Nives Pećina-Šlaus
